# Dynamic Gaming Case of the R-Interdiction Median Problem with Fortification and an MILP-Based Solution Approach

**Yiyong Xiao**, **Pei Yang, Siyue Zhang**, **Shenghan Zhou** *, **Wenbing Chang and Yue Zhang**

School of Reliability and System Engineering, Beihang University, Beijing 100191, China;
xiaoyiyong@buaa.edu.cn (Y.X.); yangpei@buaa.edu.cn (P.Y.); zhang_sy@buaa.edu.cn (S.Z.);
changwenbing@buaa.edu.cn (W.C.); Zhangyue1127@buaa.edu.cn (Y.Z.)
* Correspondence: zhoush@buaa.edu.cn; Tel.: +86-010-82316003

**Abstract:** This paper studies the cyclic dynamic gaming case of the *r*-interdiction median problem with fortification (CDGC-RIMF), which is important for strengthening a facility's reliability and invulnerability under various possible attacks. We formulated the CDGC-RIMF as a bi-objective mixed-integer linear programming (MILP) model with two opposing goals to minimize/maximize the loss from both the designer (leader) and attacker (follower) sides. The first goal was to identify the most cost-effective plan to build and fortify the facility considering minimum loss, whereas the attacker followed the designer to seek the most destructive way of attacking to cause maximum loss. We found that the two sides could not reach a static equilibrium with a single pair of confrontational plans in an ordinary case, but were able to reach a dynamically cyclic equilibrium when the plan involved multiple pairs. The proposed bi-objective model aimed to discover the optimal cyclic plans for both sides to reach a dynamic equilibrium. To solve this problem, we first started from the designer's side with a design and fortification plan, and then the attacker was able to generate their worst attack plan based on that design. After that, the designer changed their plan again based on the attacker's plan in order to minimize loss, and the attacker correspondingly modified their plan to achieve maximum loss. This game looped until, finally, a cyclic equilibrium was reached. This equilibrium was deemed to be optimal for both sides because there was always more loss for either side if they left the equilibrium first. This game falls into the subgame of a perfect Nash equilibrium—a kind of complete game. The proposed bi-objective model was directly solved by the CPLEX solver to achieve optimal solutions for small-sized problems and near-optimal feasible solutions for larger-sized problems. Furthermore, for large-scale problems, we developed a heuristic algorithm that implemented dynamic iterative partial optimization alongside MILP (DIPO-MILP), which showed better performance compared with the CPLEX solver when solving large-scale problems.

**Keywords:** facility location; *r*-interdiction median problem with fortification (RIMF); cyclic dynamic equilibrium game; mixed-integer linear programming (MILP)

---

## 1. Introduction

The reliability and security of critical infrastructures is vital to the stability of a country's social system. The growing threat of international terrorism may imperil facilities and cause massive loss to people's lives and properties. Such man-made disasters often seriously damage facilities, resulting in general or partial impacts on the normal order of society, leading to large-scale supply shortages of products or service interruptions. Some attacks will affect the sustainability of key facilities, making it difficult to ensure the normal supply, so it is necessary to consider the location of facilities under

attack. In practice, there may be multiple rounds of confrontation between the attackers and defenders, so dynamic attack defense issues need to be considered to ensure the long-term sustainability of the facilities.

Terrorists always seek out the most critical facilities to conduct attacks and maximize the loss of the social system. To strengthen the security and reliability of critical facilities, facility locations should be well planned and fortified, and coping measures, such as backup systems and service reassignments, should be prepared for in advance of an attack. Our problems fall into the class of an incapacitated facility location, which is a variant of the *p*-median problem first proposed by German scholar Alfred Weber [1]. In the paper of Huizhen [2], they further study the mathematical nature of the semiLagrangian relaxation (SLR) applied to solve the un-capacitated facility location (UFL). On this basis, SLR for UFL is improved in theory, and the way to improve its computing capability is discussed. Monabbati [3] use their proposed sub-additive dual ascent procedure to find an optimal sub-additive dual function based on Klabjan's generator sub-additive function to solve the so called uncapacitated facility location problem (UFLP). Glover [4] proposes a simple multi-wave algorithm for solving the uncapacitated facility location problem (UFLP) to minimize the combined costs of selecting facilities to be opened and assigning each customer to an opened facility in order to meet the customers' demands. On the basis of an incapacitated facility location problem with interdiction and fortification, the *r*-interdiction median problem with fortification (RIMF) occurs. In the existing literature, node interdictions have been divided into two groups, i.e., *r*-interdiction median models (RIM) and r-interdiction covering models (RIC). The RIM model was originally formulated by Church and Scaparra [5], where interdiction strategies seek *p* facilities that have already been located. The model determines a subset of *r* facilities, and if these facilities are lost, the impact on the average service distance or the total weighted distance from the customer is at its greatest. In the RIC model, the objective is to specify *r* facilities of *p* existing facilities, which, when removed, results in a maximum drop in coverage. Church and Scaparra [6] then put forward the interdiction median problem with fortification (IMF) model, which involves finding a subset of *q* facilities in *p* different locations of the supply or service facility; when these *q* facilities harden, they provide the best protection from a subsequent optimal *r*-interdiction strike. Church and Scaparra [7] propose the *r*-interdiction median problem with fortification (RIMF) model, where the leader and follower respectively conduct interdiction and fortification, assuming that one side has complete information about the other side. Figure 1 below shows the study process from the *p*-median problem to the RIMF problem.

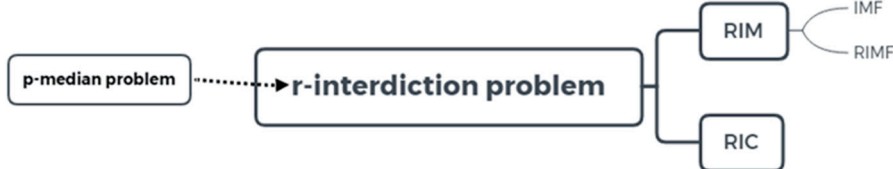

**Figure 1.** The study process from the *p*-median problem to the *r*-interdiction median problem with fortification (RIMF) problem.

A number of studies on the *r*-interdiction problem in the literature have strived to develop more reliable and invulnerable systems. Snyder [8] presents a stochastic location model with risk pooling (LMRP) to optimize location, inventory, and allocation decisions under random parameters described by discrete scenes. Scaparra et al. [9] present an optimization modeling approach to allocate protection resources among a system of facilities so that the disruptive effects of possible intentional attacks to the system are minimized. In the work of Aksen et al. [10], the authors elaborate on a budget-constrained extension of the RIMF model, where the objective function is to find the optimal allocation of protection resources to a given service system consisting of *p* facilities such that the disruptive effects of *r* possible attacks are minimized. Liberatore et al. [11] present the stochastic *r*-interdiction median problem with fortification (S-RIMF), where the model takes into consideration a random number of

possible losses. Zhu et al. [12] studies the *r*-interdiction median problem with probabilistic protection, assuming that defensive resources are allocated based on the degree of importance of the service facility. Medal et al. [13] integrate the facility location and the facility reinforcement decisions into the formulation and assume that the decision-maker is risk averse and thus interested in mitigating against the facility disruption scenario with the greatest consequences. Mahmoodjanloo [14] puts forward a tri-level model under the defender–attacker–defender framework based on leader–follower games for facility protection against disturbance in an *r*-interdiction median problem. Mahmoodjanloo [14] focuses on reducing the effect of intentional attacks, in which facilities are located and strengthened within a limited budget. Sadeghi [15] presents a new formulation and solution method for the partial fortification and interdiction of a tri-level shortest path problem which extends the existing network interdiction models to a more practical environment. Zheng et al. [16] present an exact approach to solve the *r*-interdiction median problem with fortification. Their methods include a greedy heuristic and an iterative algorithm, solving a set cover problem iteratively to ensure the best solution upon termination. Khanduzi [17] implements two novel approaches to solve the multi-period interdiction problem with fortification. Roboredo [18] proposes a branch-and-cut algorithm for the RIMF problem, which is the best exact algorithm found. Furthermore, Roboredo [19] also put forward a branch-and-cut algorithm for the *r*-interdiction covering problem with fortification (RICF) problem, which is faster in solving large instances compared with the exact method found in the literature. Biswas and Pal [20] propose an interesting hybrid goal programing model and genetic algorithm in a fuzzy environment. Barma et al. [21] present a novel linear programming(LP) model with antlion optimization algorithm for multi-depot vehicle routing problem(VRP). In conclusion, in the previous literatures, there is almost no multi-round attack defense confrontation. Most papers generally consider a bi-objective function for at most a three level RIMF problem. In this paper we present, for the first time, a cyclic equilibrium gaming case of the *r*-interdiction median problem with fortification based on the study of Dong [22]. We consider it a dynamic Stackelberg game [23], involving two non-cooperative, fully rational players to protect (or attack) the facility as much as possible. Each side makes the most optimal decision based on the other's decision. We present a bi-objective mixed integer linear programming model to formulate the cyclic dynamic gaming case of the *r*-interdiction median problem with fortification (CDGC-RIMF). Using the cyclic algorithm, the computer generate two decision packages for both the attacker and defender until the equilibrium was cyclically reached. For the large-sized problem, we developed a heuristic algorithm based on the partial optimization strategy [24,25] to efficiently solve the problem with near-optimal solutions. Contributions of the paper are outlined as follows.

- This article first brings up the CDGC-RIMF problem, which considers cyclic decision periods/phases in large-scale scenes with two opposite objectives.
- This article constructs the mixed-integer linear programming (MILP) model so as to solve it by an efficient and easy way through the CPLEX solver.
- For the large-size scenes, we developed a heuristic algorithm that implemented dynamic iterative partial optimization alongside MILP (DIPO-MILP).
- Our model put the wise decision of the attacker into consideration. The objective of attacker was to consider the optimized rearrangement of the defender faced with the destruction to achieve the worst loss possible.
- Our algorithm generated two decision packages for both the attacker and defender, thereby reaching an optimal cyclic equilibrium.
- In the cyclic confrontation process, each side made the most optimal confrontation plan until the balance of the cyclic confrontation was reached. Through this model, the decision package of the two sides could be produced. For the protector, it could predict the other side's decision in advance (the protector knew that the attacker would make the most optimal confrontation plan) and then make the decision for the next step. It could help the facility manager make wise

decisions for site selection and defense in advance, so as to ensure the long-term sustainability of the facility.

The remaining paper is organized as follows. In Section 2, we formally describe the CDGC-RIMF model. In Section 3, we present the notation, the property of the cyclic bi-objective MILP model, and a partial optimization algorithm for large-scale problems. In Section 4, we present some computational experiments, including the small- and large-scale problems, and analyze the protection effect to make up a fortified facility. Finally, in Section 5, we summarize our work and present suggestions for future works.

## 2. Problem Description and Formulation

The problem involved a general service supply system composed of several facilities and a number of demand nodes that received goods or services from their nearest service sites. A number $p$ of facilities were built out of a set of potential locations sites, denoted as $F$ ($j \in F$). Notation $N$, indexed by $i$, represents the set of demand nodes. Each demand node $i$, $i \in N$ was associated with a demand $a_i$. The distances between the facilities and the demand nodes were known parameters, denoted as $d_{ij}$, where $i \in N$ and $j \in F$. The service cost was expressed as the sum of total demands weighted by the distances to their closest facilities. Each candidate site was characterized with a fixed cost $f_j$ to build a facility and a fixed fortification cost $g_j$ to protect the built facility. The attacker (follower) intended to carry out the most devastating attack possible and was assumed to have the ability to destroy the maximum $r$ number of facilities after considering all possible design plans from the designer. The designer, who was able to protect $g$ facilities from being attacked, tried to minimize the establishment and protective cost as well as system loss with consideration of all possible attacks by the attacker.

We formulated the cyclic dynamic gaming case of the $r$-interdiction median problem as a bi-objective MILP model, as shown below. The sets, parameters, and decision variables used in the mathematical formulation were introduced in advance, as follows.

*Sets:*
$N$: Set of demands;
$F$: Set of candidate sites.
*Parameters:*
$i$: Index of demand that $i \in N$;
$j$: Index representing candidate site that $j \in F$;
$a_i$: Demand of node $i$;
$m$: Number of potential sites;
$p$: Number of built (located) facilities;
$r$: Number of interdicted facilities;
$q$: Number of defense facilities, q < p, r < p–q;
$g_j$: Setup cost at site $j$;
$h_j$: Fortification cost at site $j$;
$d_{ij}$: Distance from demand $i$ to site $j$;
$M$: A large number.
*Variables:*
$x_j$: $\begin{cases} 1, \text{ if site j is selected to operate a facility} \\ 0, \text{ otherwise} \end{cases}$ ;

$y_{ij}$: $\begin{cases} 1, \text{ if customer i is served by the facility at site j} \\ 0, \text{ otherwise} \end{cases}$ ;

$s_j$: $\begin{cases} 1, \text{ if the facility at site i is interdicted} \\ 0, \text{ otherwise} \end{cases}$ ;

$z_j$: $\begin{cases} 1, \text{ if the facility at site j is fortified} \\ 0, \text{ otherwise} \end{cases}$ ;

$b_j$: $\begin{cases} 1, \text{ if sit } j \text{ has a valid facility} \\ 0, \text{otherwise} \end{cases}$ ;

$X$: Set for facility location, $X = \{x_j\}$;

$Y$: Set for service assignment, $Y = \{y_{ij}\}$;

$S$: Set for interdicted facility, $S = \{s_j\}$;

$Z$: Set for defense facility, $Z = \{z_j\}$.

The main goal of the designer was to design a service network with the objective of minimizing the sum of fixed and variable costs of a system facing disturbance. The goal of the attacker was to maximize the objective of the designer.

Objective 1 (from the designer):

$$\text{Minimize}: \min f(X, Z) = \sum_{i \in N, j \in F} a_i d_{ij} y_{ij} + \sum_{j \in F} (g_j x_j + h_j z_j). \tag{1}$$

Objective 1 (from the attacker):

$$\text{Maximize}: \max f(S) = \sum_{i \in N, j \in F} a_i d_{ij} y_{ij}. \tag{2}$$

Subject to:

(1) $\sum_{j \in F} x_j \leq p$　　　　$\forall j \in F,$

(2) $\sum_{j \in F} s_j \leq r$　　　　$\forall j \in F,$

(3) $\sum_{j \in F} z_j \leq q$　　　　$\forall j \in F,$

(4) $\sum_{j \in F} y_{ij} = 1$　　　　$\forall i \in N,$

(5) $\begin{cases} y_{ij} \leq x_j \\ y_{ij} \leq 1 - s_j + z_j \end{cases}$　　　$\forall i \in N, j \in F,$

(6) $z_j \leq x_j$　　　$\forall j \in F,$

(7) $\begin{cases} b_j \leq x_j \\ b_j \leq 1 - s_j + z_j \\ b_j \geq x_j - s_j \\ b_j \geq x_j + z_j - 1 \end{cases}$　　　$\forall j \in F,$

(8) $y_{ij} \cdot d_{ij} \leq d_{ij'} + M \cdot (1 - b_{j'})$　　　$\forall i \in N; j, j' \in F,$

(9) $x_j \in \{0, 1\}, y_{ij} \in \{0, 1\}, z_j \in \{0, 1\}, s_J \in \{0, 1\}, b_j \in \{0, 1\}$　　　$\forall i \in N, j \in F.$

In the above model, Constraint (1) indicates that the designer built $p$ facilities out of the set of candidate facilities. Constraint (2) ensures that, at most, $r$ facilities were to be interdicted. Constraint (3) stipulates that the designer could maximally protect $q$ facilities. Constraint (4) makes sure that each demand was serviced by one facility. Constraint (5) states that only the facilities that were set up and not attacked at the same time (or were attacked but protected at the same time) could serve the demands. Constraint (6) indicates that only the built facilities were allowed to be protected. Constraint (7) determines whether a site had a valid facility, i.e., $b_j = 1$ if site $j$ had a valid facility or $b_j = 0$ if otherwise. Constraint (8) is a new constraint for the closest assignment, a different version from the related constraints devised by Church [6] and Liberatore et al. [11]. This constraint normally used the set of existing valid facilities (not including $j$) that was further than $j$ from demand $i$. However, these sets had to be recalculated when the fortification or interdiction changed; a phenomenon associated with considerable computational complexity. Constraint (9) states that all of the decision variables were binary.

### 3. Solution Approaches for Cyclic Dynamic Equilibrium

*3.1. Solution Definitions and Analysis*

Below are the notations used for the definitions and analysis.

U: Set of all possible design plans;
V: Set of all possible attack plans;
D: A design plan and $D \in U$;
A: An attack plan and $A \in V$;
A|D: The optimal attack plan for the given design plan;
D|A: The optimal design plan for the given attack plan;
$O_{D/A}$: The optimal minimum objective cost of the design plan for the given attack plan;
$O_{A/D}$: The optimal maximum objective cost of the attack plan for the given design plan.

**Definition 1.** *General optimal design (GOD) plan: The general optimal design plan was a plan by the designer that was optimized for a given attack plan.*

(1)   *ATTACKER: A= {$A_i$/ $A_i \in V$};*
(2)   *DESIGNER: D = {$D_i$/$D_i$ satisfies $O_{D/A}$, $A_i \in V$, $D_i \in U$}.*

When solving the objective of the designer, we fixed the variable $s_j$ in S with given values, then searched for the minimum cost of all of the designer's plans to thwart the opponent, as Figure 2 shows.

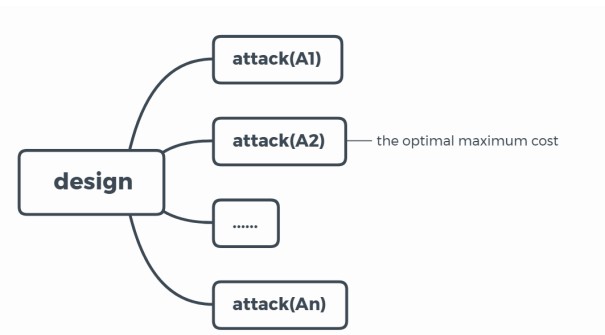

**Figure 2.** The optimal attack strategy.

**Definition 2.** *General optimal attack (GOA) plan: The general optimal attack was the plan by the designer that was optimized for a given design plan.*

(1)   *DESIGNER: D = {$D_i$/ $D_i \in U$};*
(2)   *ATTACKER: A = {$A_i$/$A_i$ satisfies $O_{A/D}$, $A_i \in V$, $D_i \in U$}.*

When solving the objective of the attacker, we fixed the variable $x_j, z_j$ in S with given values, then searched for the maximum cost of all of the attacker's plans to thwart the opponent, as Figure 3 shows.

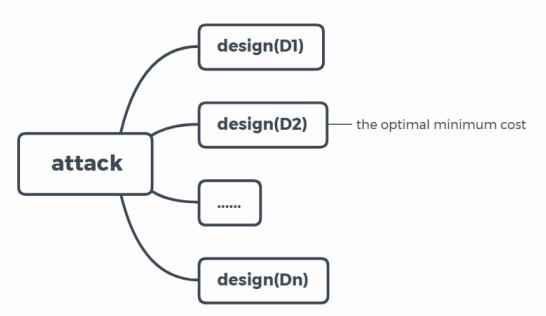

**Figure 3.** The optimal design strategy.

**Definition 3.** *The ideal design plan (IDP): The ideal design plan had the lowest objective cost of all of the design plans, occurring when there was no interdiction. The ideal design plan was a special case of GOD.*

(1)     *ATTACKER: A= { θ};*
(2)     *DESIGNER: D = {D_i/D_i satisfies $O_{D/A}$, $A_i \in V$, $D_i \in U$}.*

*In this situation of the fortification set Z= {θ}, it was obvious that the designer did not need to set up the fortification facility because there was no interdiction from the attacker. The ideal design plan was a special case of GOD.*

**Definition 4.** *The ideal attack plan (IAP): The ideal attack plan had the highest objective cost of all of the attack plans, occurring when there was no fortification. The ideal attack plan was a special case of GOA.*

(1)     *DESIGNER: D = {D_i/ $D_i \in U$};*
(2)     *ATTACERK: A = {A_i/A_i satisfies $O_{A/D}$, $A_i \in V$, $D_i \in U$}.*

*This situation, where Z = {θ}, was very useful for the attacker. Attacking the facility unprotected would lead to huge damage.*

**Definition 5.** *The cycled confrontation strategy (CCS): The cycled confrontation strategy was expressed as {A1/D1, D1/A2, A2/D3, D3/A4, A4/D1}. There were two sets of plans from two sides, where each plan of one set was the optimal plan when responding to the opposite plan from the other side. The cycle stopped when the end plan was the same as the existing plan, as shown in Figure 4.*

**Definition 6.** *The round of confrontation (RC): Every time one side carried out a strategy, the other side found a strategy to use to fight back. This was one round of confrontation. In the example shown in Figure 4, we demonstrate that there were nine rounds from the beginning to the end of the opposite strategies.*

**Definition 7.** *The round of cycled confrontation (RCC): In the example shown in Figure 4, we demonstrate that there were six rounds from the beginning of the cycle (D2) to the end of the cycle when D2 existed again.*

**Definition 8.** *The evaluation indicators for two kinds of solution methods, as shown in Table 1.*

**Table 1.** The evaluation indicators.

| | Index | Name |
|---|---|---|
| 1 | $\overline{C_A}$ | Average attack cost |
| 2 | $\overline{C_D}$ | Average design cost |
| 3 | $\overline{C_A}–\overline{C_D}$ | The optimality of solutions |

**Property 1.** *The cycled confrontation strategy began with the IDP and IAP.*

**Property 2.** *The cycled confrontation strategy existed because the plans from both sides were finite.*

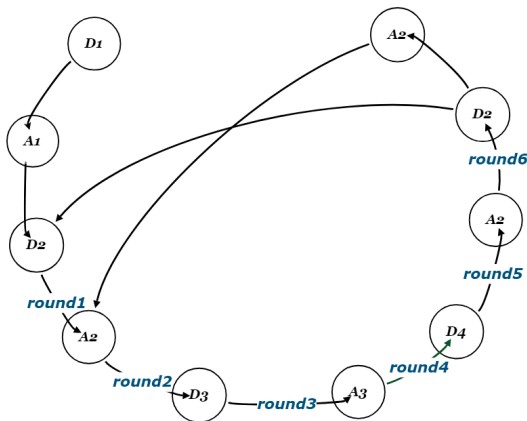

**Figure 4.** The optimal attack strategy.

### 3.2. A Bi-Objective Solution Framework

We solved CDGC-RIMF with a MILP-based solution approach. Our approach started by creating an initial design plan D, then fixed $D_i$ (fix $x_i$, $z_i$) to achieve the attack plan $A_i$ (fix $s_i$) using the MIP solver CPLEX. After the attack plan, we fixed the attack plan $A_i$ (fix $s_i$) to get the next design plan. The solution process continued until the repeated attack plan existed, as shown in Figure 5.

---

1) Let i = 1.

2) Find an initial design plan $D$ , and let $D_i \leftarrow D$

3) Repeat

     4) Fix $D_i$ as a given design plan and use the MIP solver to find am

         attack plan, noted as $A_i$ , forming $A_i // D_i$

        **Partial Optimization Loop**

     5) Fix $A_i$ as a given attack plan and use the MIP solver to find a

         design plan, noted as $D_{i+1}$ , forming $D_{i+1} // A_i$

        **Partial Optimization Loop**

     6) Let i = i + 1

7) Until $D_i = D_{ii}$ exists where $1 \leq D_{ii} \leq i-1$

8) Output the equilibrium method

---

**Figure 5.** The bi-objective solution approach.

### 3.3. A Heuristic Partial Optimization Algorithm

The large-sized complex problem with many possible decision variables could not be optimized completely within an acceptable Central Processing Unit (CPU) time. Therefore, some heuristic algorithms were put forward to reduce the solution space so as to save time and improve the calculation efficiency, in particular, the heuristic concentration algorithm. As Rosing and Hodgson [26] illustrated in their paper, the heuristic concentration procedure mainly consists of two stages. Firstly, it was necessary to find a concentration set according to some rules, and secondly, the optimal solution in the concentration set could be solved every time by generating a concentration set. Based on Rosing and Hodgson's heuristic concentration procedure, the MILP-based neighborhood searching algorithms by Xiao et al. [24,25] and You et al. [27,28] were introduced, and the concept of the partial set was also presented. For this paper, we developed a heuristic algorithm for large-sized problems following

the steps showed above. First, we fixed all of the nodes, then generated iteratively different partial sets that were subsequently used to unfix the nodes in the partial set, and then optimized the unfixed nodes. A rule used to generate the partial set was that nodes selected to be in the partial set more times should have a lower possibility of being selected in the next round. The main framework of the proposed heuristic algorithm is outlined in Figure 6. The partial optimization forced the optimization to stay within only a small range of the variables so it could be achieved in a short computational time. An iterative implement of partial optimization on different selected ranges continuously improved the solution until it was good enough or until the time limit was reached.

As shown in Figure 6, the partial optimization algorithm (POA) procedure started with an initial random attacker plan $(x_0, z_0)$ and a designer plan $(s_0)$, obtained by randomly generating the number of $q$ interdictions and $p$ facilities. Then (in Step 1,2), the confrontation cyclic started with the initial attacker and the designer plan. The first confrontation plan was from the defender. The index $P$ suggested an improvement for every round of partial optimization. The parameter $W$ indicated the number of to-be-selected nodes for partial optimization and was assigned an initial value in Step 3. Our experiments showed that parameter $W$ was sensitive and considerably influenced the algorithm efficiency. The appropriate value interval for $W$ was between 35 and 45, according to our experimental comparisons. After that, the partial optimization loop started at Step 4, containing Steps 4.1–4.11. In Step 4.2–4.5, all variable instances of $x_j$ were first fixed, and then a number $W$ of instances were selected to be unfixed. Each time, we selected a number $W$ of $x_j$ following the selection rule, that nodes chosen more often had a lower possibility of being chosen. After that, the CPLEX solver was applied to optimize the selected nodes (in Step 4.6). In Steps 4.7 and 4.8, the new solution was judged regarding whether it improved upon the incumbent solution. If yes, then it was accepted as the new incumbent solution and the non-improvement index $P$ was reset to 0; otherwise, the $P$ was increased by one. The loop stopped if there was no further improvement even after $P_{max}$ continuous attempts, or if the total CPU time was longer than the set limited time $T_{max}$, or if all of the nodes were optimized at the same time (in Step 4.6). In steps 4.10–4.12, the number $W$ was adjusted dynamically to ensure that the solution time solved through CPLEX was between $t_{min}$ and $t_{max}$. The number $W$ increased by 5 if the last CPU time was less than $t_{min}$ or reduced by 5 ($\Delta = 5$) if the last CPU time was more than $t_{max}$. Finally, the obtained $C$ was the best solution found and became the output of the algorithm. The attacker adopted similar partial optimization as the partial optimization in the defender's plan to develop its coping plan.

---

Procedure: The partial optimization algorithm (POA)

Input parameters: $T_{max}$, $P_{max}$, $t_{min}$, $t_{max}$, $W_{min}$, $\Delta$, $W_0$; output result: $x_j$, $z_j$, $s_j$, *a.obj*, *d.obj*

1) For the designer: Let $x_j$, $z_j \leftarrow$ Initialization: Random generation of a designer strategy

2) For the attacker: Let $s_j \leftarrow$ Initialization: Random generation of an attacker strategy

3) Let $x_j$, $z_j$, $s_j \leftarrow x_0$, $z_0$, $s_0$, $P \leftarrow 0$, and $W \leftarrow W_0$

4) **Iterative Neighborhood Search (INS) Loop Begins**

4.1) **For the designer:**

4.2) Let $f \leftarrow$ null

4.3) Apply a rule to select a number $W$ of nodes from $F$ into $f$

4.4) For $j$ in $F$, fix all $x_j$ decision variables

**Figure 6.** *Cont.*

> 4.5) For *j* in *f*, unfix $x_j$ decision variables
>
> 4.6) Call the CPLEX solver to get a new solution $X'$,$Z'$, d.obj' and record the used CPU time *t*
>
> 4.7) **If** *d.obj'* improved upon *d.obj*, **then** let *d.obj* ← *d.obj'* and $P \leftarrow 0$
>
> 4.8) **Otherwise,** let $P \leftarrow P + 1$
>
> 4.9) **If** $P \geq P_{max}$ **or** total CPU time $\geq T_{max}$ or $W \geq$ number of CFs **then** break the loop and stop
>
> 4.10) **If** $t \leq t_{min}$, **then** let $W \leftarrow W + \Delta$
>
> 4.11) **if** $t \geq t_{max}$, **then** let $W \leftarrow W - \Delta$
>
> 4.12) **if** $W < W_{min}$, **then** let $W \leftarrow W_{min}$
>
> 4.13) **For the attacker: The process was similar to 4.2 – 4.12:**
>
>       **Just substitute $x_j$ with $s_j$ and substitute *d.obj* with *a.obj***
>
> 4) **Loop End**

**Figure 6.** The framework of the POA algorithm.

## 4. Computational Tests

The goal of the computational experiments was to validate the model. The codes by AMPL were run on a Linux PC server with two 2.30 GHz Intel Xeon (R) CPUs and 128 GB RAM. We used the MIP solver AMPL/CPLEX (version 12.6.0.1) to solve the tested instances.

### 4.1. The Generation of the Tested Instances

The data sets and the parameter settings used in the experiments were generated randomly, as shown in Table 2. In this paper, we used small-, medium-, and large-scale problems to validate our model.

**Table 2.** The parameters for different scale problems.

|  |  | Nodes of Demands CU | Candidate Facilities CF | Facilities to be Built F(*p*) | Number of Attacks (*r*) | Number of Fortification *z*(*q*) |
|---|---|---|---|---|---|---|
| **Small-Scale** | **1** | 25 | 10 | 5 | 3 | 1 |
|  | **2** | 50 | 20 | 10 | 5 | 3 |
|  | **3** | 75 | 30 | 15 | 8 | 4 |
|  | **4** | 100 | 40 | 20 | 10 | 5 |
| **Medium-Scale** | **5** | 250 | 50 | 25 | 13 | 6 |
|  | **6** | 300 | 60 | 30 | 15 | 8 |
|  | **7** | 350 | 70 | 35 | 18 | 9 |
|  | **8** | 400 | 80 | 40 | 20 | 10 |
| **Large-Scale** | **9** | 900 | 90 | 45 | 23 | 11 |
|  | **10** | 1000 | 100 | 50 | 25 | 13 |
|  | **11** | 1100 | 110 | 55 | 28 | 14 |
|  | **12** | 1200 | 120 | 60 | 30 | 15 |

Note: candidate facility (CF); client (CU).

For the small-scale problem, we set a number of 10, 20, 30, or 40 candidate facilities (CFs), respectively, within a square region of $100 \times 100$ distance units. The actual limited number of facilities (P) was set by CF/2. Every selected facility served five clients, so the clients (CU) assigned to each facility were about 25, 50, 75, or 100. The number of fortified facilities and interdiction facilities were as

shown in column 6–7. The number of fortified facilities ($r$) was calculated by $p/2$, while the number of interdiction facilities was calculated by $r/2$. The coordinates of the facilities and the clients in the small-scale problem were generated between 0 and 100. For the medium problem, the coordinates of the facilities and clients were in the square region of $200 \times 200$ distance units, with the CFs listed as 50, 60, 70, or 80. The number of CUs, $r$s, and $q$s were calculated in a similar way, as in the small-scale problem. In the large-scale problem, these values were in the square region of $400 \times 400$ distance units. The numbers relating to CF, $p$, CU, $r$, and $q$ are shown in columns 3–7.

### 4.2. Analysis of the Tested Results

Tables 3 and 4 display the computation time, RC, and RCC. The first six columns show the parameters CF, F, CU, r, and q. The seventh and eighth columns, respectively, contain cumulative computing times and independent run times. The last two rows display the round of confrontation and the round of cycled confrontation. As shown in Table 3, ($r > q$), RCC 4 remained unchanged as the CF increased, while Table 3 ($r < q$) shows that there was a different RCC 6 for the 50 CFs. The solution time increased as the parameters of CF, F, CU, r, and q increased. For the small- and medium-scale problems, calculations were finished in 500 s. However, it took a long time to solve the large-scale problem; therefore, it is necessary to solve the large-scale problem using another, more efficient, approach.

**Table 3.** The results for small- and medium-scale problems using CPLEX ($r > q$). Round of confrontation (RC); round of cycled confrontation (RCC); candidate facility (CF); client (CU).

|  | CF | F(p) | CU | s(r) | z(q) | time(s) | RC | RCC |
|---|---|---|---|---|---|---|---|---|
| **1** | 10 | 5 | 25 | 3 | 1 | 0 | 5 | 4 |
| **2** | 20 | 10 | 50 | 5 | 3 | 1 | 9 | 4 |
| **3** | 30 | 15 | 75 | 8 | 4 | 3 | 9 | 4 |
| **4** | 40 | 20 | 100 | 10 | 5 | 5 | 5 | 4 |
| **5** | 50 | 25 | 250 | 13 | 6 | 62 | 7 | 4 |
| **6** | 60 | 30 | 300 | 15 | 8 | 170 | 9 | 4 |
| **7** | 70 | 35 | 350 | 18 | 9 | 267 | 7 | 4 |
| **8** | 80 | 40 | 400 | 20 | 10 | 527 | 7 | 4 |
| **9** | 90 | 45 | 900 | 23 | 11 | 5019 | 11 | 4 |
| **10** | 100 | 50 | 1000 | 25 | 13 | 6297 | 9 | 4 |
| **11** | 110 | 55 | 1100 | 28 | 14 | 12,030 | 11 | 4 |
| **12** | 120 | 60 | 1200 | 30 | 15 | 14,687 | 9 | 4 |

**Table 4.** The results for the small- and medium-scale problems ($q > r$).

|  | CF | F(p) | CU | s(r) | z(q) | time(s) | RC | RCC |
|---|---|---|---|---|---|---|---|---|
| **1** | 10 | 5 | 25 | 1 | 3 | 0 | 5 | 4 |
| **2** | 20 | 10 | 50 | 3 | 5 | 1 | 7 | 4 |
| **3** | 30 | 15 | 75 | 4 | 8 | 2 | 7 | 4 |
| **4** | 40 | 20 | 100 | 5 | 10 | 8 | 7 | 4 |
| **5** | 50 | 25 | 250 | 6 | 13 | 84 | 9 | 6 |
| **6** | 60 | 30 | 300 | 8 | 15 | 145 | 7 | 4 |
| **7** | 70 | 35 | 350 | 9 | 18 | 289 | 7 | 4 |
| **8** | 80 | 40 | 400 | 10 | 20 | 530 | 7 | 4 |
| **9** | 90 | 45 | 900 | 11 | 23 | 3379 | 7 | 4 |
| **10** | 100 | 50 | 1000 | 13 | 25 | 6219 | 9 | 4 |
| **11** | 110 | 55 | 1100 | 14 | 28 | 7219 | 7 | 4 |
| **12** | 120 | 60 | 1200 | 15 | 30 | 141,119 | 11 | 4 |

We chose one example to illustrate the concrete game process between the two sides. We took the $10 \times 25$ problem as an example to demonstrate the confrontation process. As shown in Table 5, the design plans, attack plans, and the cost were listed until the same attack plan existed. Figure 7a–f gives the model solution of Table 5 in graphical form.

**Table 5.** The strategy of both sides (for data: $10 \times 25$, $p = 6$, $r = 3$, $q = 2$).

| Decision Maker | | Designer | | Attacker | Cost |
|---|---|---|---|---|---|
| | | Location | Fortification | | |
| 1 | design | 1,5,8,10 | | | 47,251 |
| 1 | attack | 1,5,8,10 | | 1,5,10 | 89,291 |
| 2 | design | 3,4,5 | 5 | 1,5,10 | 50,119 |
| 2 | attack | 3,4,5 | 5 | 3,4 | 76,437 |
| 3 | design | 1,5,8,10 | | 3,4 | 47,251 |
| 3 | attack | 1,5,8,10 | | 1,5,10 | 89,291 |

As shown in Figure 7a–f, points selected to be used for facilities were colored green and the outline of the fortified facilities was black. The lines in each figure described assignments of demand to selected facilities. The attack points were displayed as red triangles. In the first stage, when $i = 1$, the designer selected P (P = 5) facilities from potential locations considering the minimum weighted distance. Faced with the designer's strategy, the attacker adopted an optimal method to ensure maximum destruction. In the second stage, when $i = 2$, the designer decided how to protect and establish the most reliable facilities to reduce loss. Every time when the attackers carried out the attack strategy, this model provided the protector with a most effective way to maintain the supply as much as possible. Wise decisions helped the facility designer to keep the facilities sustainable and reliable against the multi-confrontation between two sides. The confrontation process continued until the same attack plan existed., and then the final equilibrium was reached.

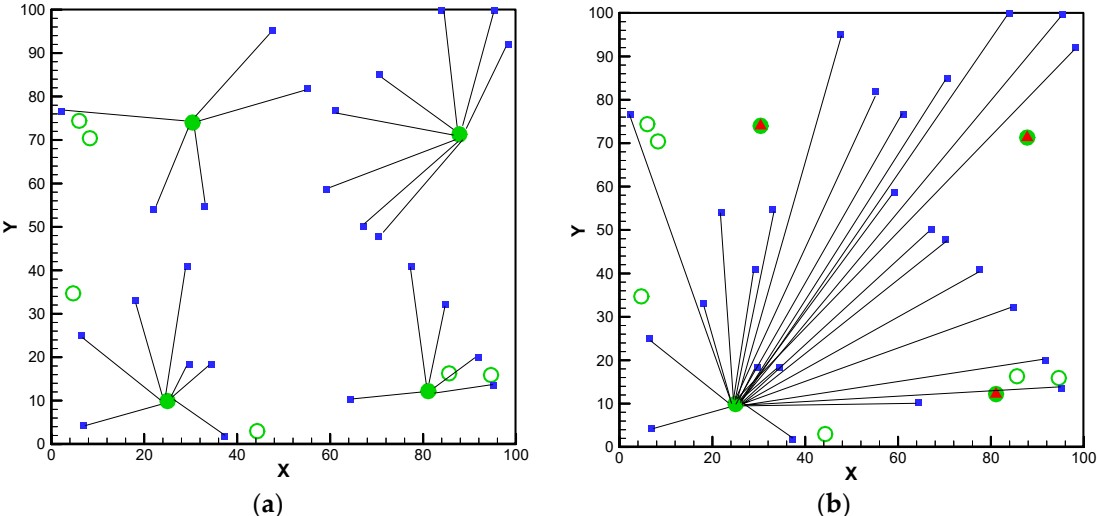

**Figure 7.** *Cont.*

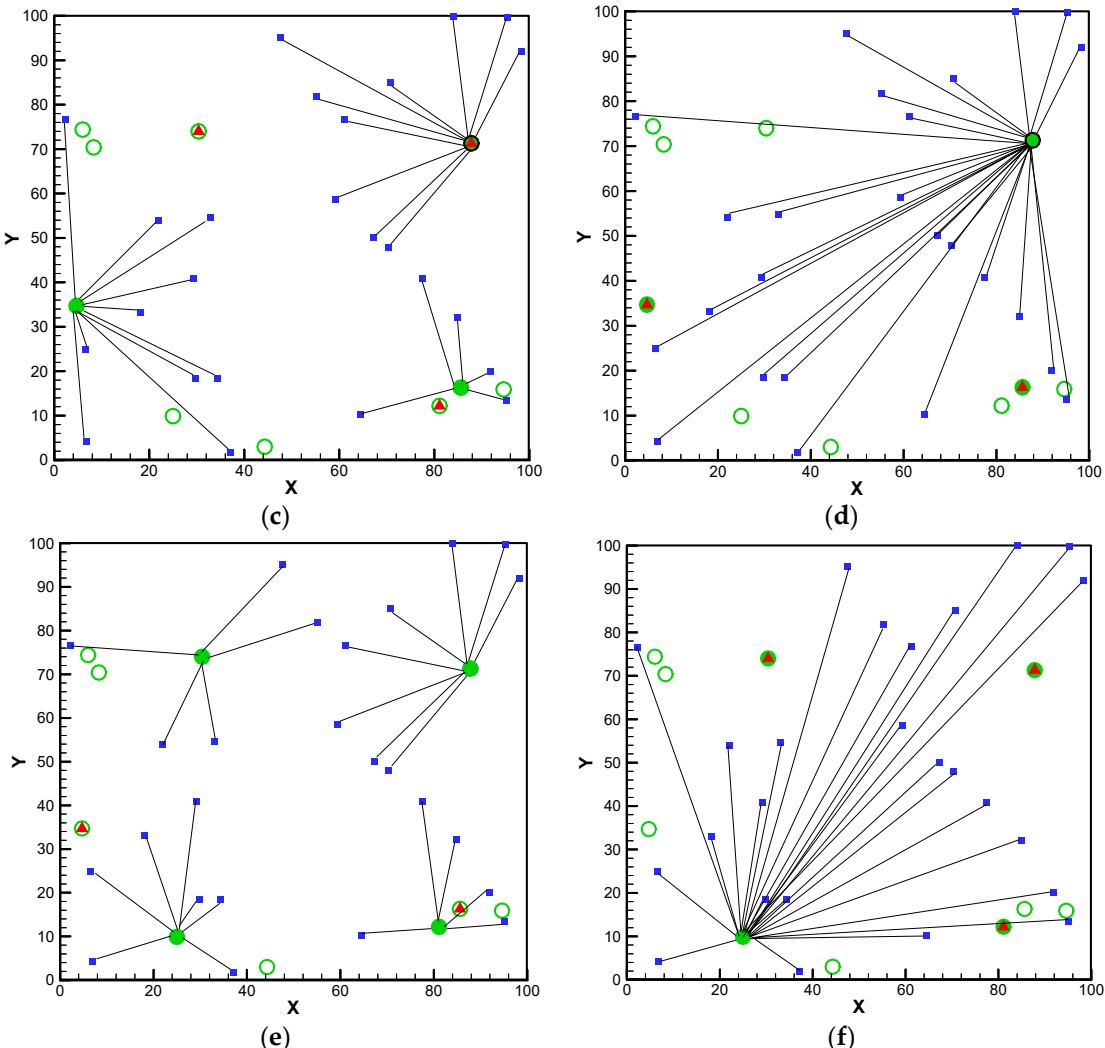

**Figure 7.** (**a**) $i = 1$, the optimal designer plan; (**b**) $i = 1$, the optimal attacker plan; (**c**) $i = 2$, the optimal designer plan; (**d**) $i = 2$, the optimal attacker plan; (**e**) $i = 3$, the optimal designer plan; and (**f**) $i = 3$, the optimal designer plan. ⭕—the potential locations of facilities; 🔺—interdiction on the facilities; 🟢—facilities in use; ⬤—fortified facilities in use, 🟦—demands.

For the large-scale problem, we implemented the POA and CPLEX to solve the model at sizes of $90 \times 900$, $100 \times 1000$ and $110 \times 1100$. This experiment showed that the POA had good performance when solving large-scale instances. As shown in Table 6, the POA was much better than CPLEX in terms of CPU time, and the average CPU time of the POA was about 85.1% shorter than that of CPLEX for size $90 \times 900$, 95.9% for size $100 \times 1000$ and 96.2% for size $110 \times 1100$. These results were obtained as averages of the CPU time from algorithms in 10 replications.

**Table 6.** The comparison of POA and CPLEX for large-scale problems.)

| CF | f(p) | cu | s(r) | z(q) | CPLEX | | POA | | Time Comparison |
|---|---|---|---|---|---|---|---|---|---|
| | | | | | $\overline{C_A} - \overline{C_D}$ | Time(s) | $\overline{C_A} - \overline{C_D}$ | Time(s) | $CP_t - PO_t / CP_t$ |
| 90 | 45 | 900 | 23 | 11 | 2,360,771 | 3890 | 2,368,109 | 578 | 85.1% |
| 100 | 50 | 1000 | 25 | 13 | 2,471,750 | 8615 | 2,484,027 | 350 | 95.9% |
| 110 | 55 | 1100 | 28 | 14 | 2,883,380 | 15,374 | 2,908,162 | 581 | 96.2% |

Note: CPU time of CPLEX ($CP_t$); CPU time of POA ($PO_t$)

In our model, we considered the cycle of the confrontation strategy, so RC and RCC were not stable when we used the partial optimal algorithm to solve this problem. The RC and RCC numbers influenced the results and CPU times. Even in this situation, we found better optimal results through adjusting the parameter of the partial optimal algorithm and repeated the experiments many times. As Table 6 shows, the POA achieved better results in a shorter CPU time when CF = 90, 100, or 110.

When CF = 120, as the iteration increased, RC and RCC became even more unstable; therefore, finding better average results was more difficult. Therefore, we deleted the cycle to test the single confrontation (RC = 1); in this situation, we found that POA operated better without being influenced by the cycle confrontation. Furthermore, we performed similar experiments when CF = 90, 100, 110, and 120. The results proved that the POA performed even better in the single confrontation. When two approaches produced the same optimal results (the optimal decisions), the time taken by the POA was less than that taken by CPLEX. Among all the large-sized instances, the average CPU time of the POA was 90.3% at most and 79.7% at least, which was shorter than that of CPLEX, as shown in Table 7.

**Table 7.** The comparison of POA and CPLEX in a single confrontation (ATTACK ONCE).

| CF | f(p) | cu | s(r) | z(q) | CPLEX | | POA | | Time Comparison |
|----|------|-----|------|------|-----------|---------|-----------|---------|-------------------|
| | | | | | $\overline{C_A}$ | Time(s) | $\overline{C_A}$ | Time(s) | $CP_t$- $PO_t$/$CP_t$ |
| 90 | 45 | 900 | 23 | 11 | 4,972,729 | 370 | 4,972,729 | 75 | 79.7% |
| 100 | 50 | 1000 | 25 | 13 | 5,832,329 | 638 | 5,832,329 | 91 | 85.7% |
| 110 | 55 | 1100 | 28 | 14 | 6,437,748 | 2061 | 6,437,748 | 215 | 89.6% |
| 120 | 60 | 1200 | 30 | 15 | 6,402,308 | 1742 | 6,402,308 | 169 | 90.3% |

Note: CPU time of CPLEX ($CP_t$); CPU time of POA ($PO_t$)

We fixed the casual attack strategy, then used CPLEX and the POA to achieve the design plan faced with the same attack strategy. The POA achieved the same optimal result in less time, as Table 8 shows. The average reduced CPU time of the POA compared with CPLEX varied between 89.6% for the smallest instance (CF = 90) and 96.2% for the biggest instance (CF = 120). It was obvious that the efficiency had been improved greatly.

**Table 8.** The comparison of the POA and CPLEX in a single confrontation (DEFENCE ONCE).

| CF | f(p) | cu | s(r) | z(q) | CPLEX | | POA | | Time Comparison |
|----|------|-----|------|------|-----------|---------|------------------------|---------|-------------------|
| | | | | | $\overline{C_D}$ | Time(s) | $CP_t$- $PO_t$/ $CP_t$ | Time(s) | $CP_t$- $PO_t$/$CP_t$ |
| 90 | 45 | 900 | 23 | 11 | 1,308,912 | 1025 | 1,308,912 | 107 | 89.6% |
| 100 | 50 | 1000 | 25 | 13 | 1,480,763 | 1757 | 1,480,763 | 137 | 92.2% |
| 110 | 55 | 1100 | 28 | 14 | 1,466,200 | 2156 | 1,566,200 | 169 | 92.2% |
| 120 | 60 | 1200 | 30 | 15 | 1,586,774 | 2881 | 1,586,774 | 109 | 96.2% |

Note: CPU time of CPLEX ($CP_t$); CPU time of POA ($PO_t$)

We also discussed the effect of protective resources (q) on the designer's efficiency. Table 9 shows that the number of fortified facilities increased with the cost of changing designer. Columns 1–5 illustrate the parameters of candidate facilities, limited facilities, serving demands, number of interdiction facilities, and number of fortification facilities. For the instances CF = 10, 20, 30, and 40 listed here, the number of fortification facilities (q) varied between 1 and 10, which was no more than the number of interdiction facilities. Columns 6–7 show the RC and RCC, which almost remained the same as q increased.

**Table 9.** The sensitivity of parameter q (for the small-scale problem).

| CF | f(p) | cu | s(r) | z(q) | RC | RCC | $\overline{C_A}$ |
|----|------|----|------|------|----|-----|------|
| 10 | 5 | 25 | 3 | 1 | 6 | 4 | 48,207.00 |
|    |   |    |   | 2 ... 10 | 6 | 4 | ... |
| 20 | 5 | 25 | 3 | 1 | 10 | 4 | 63,059.00 |
|    |   |    |   | 2 | 10 | 4 | 52,318.50 |
|    |   |    |   | 3 ... 10 | ... | ... | ... |
| 30 | 15 | 75 | 8 | 1 | 8 | 4 | 85,995.00 |
|    |   |    |   | 2 | 8 | 4 | 85,189.20 |
|    |   |    |   | 3 ... 10 | ... | ... | ... |
| 40 | 20 | 100 | 10 | 1 | 6 | 4 | 101,512.33 |
|    |   |    |   | 2 | 6 | 4 | 101,354.67 |
|    |   |    |   | 3 | 6 | 4 | 101,301.67 |
|    |   |    |   | 4 ... 10 | ... | ... | ... |

Note: " ... " represents that the cost remained the same as the last iteration.

Often, most protection benefits were achieved through the first two or three fortifications, and subsequent security investments gradually reduced in efficiency. Overall, the fortification of the second facility still played an important role regarding improvement of the results. For instance, in problems where CF = 20 operating facilities, when the number of q exceeds 2, there was no improvement regarding the cost of the design.

## 5. Conclusions

In this paper, a bi-objective cyclic dynamic equilibrium gaming model of the *r*-interdiction median problem with fortification was presented. In this model, the two sides were wise to make a decision according to the operational experience. The system designer (defender) could decide regarding the interdiction. The attacker could decide to destroy essential facilities after considering the possible reassignment of the designer faced with one interdiction, which was the worst-case loss. The model could give two operational strategy packages when the two sides achieved equilibrium. To solve the large-scale problem, the partial optimization algorithm was used, and we made the comparison between the exact method using CPLEX and the partial optimization algorithm. Compared with the previous work, the strategy packages can help the facility manager to make decisions in advance against the attackers and better keep the facility in long-term sustainability. Additionally, we found that the partial optimization algorithm yielded better solutions with higher search efficiency.

In our analysis, attacks on a given facility were always successful. Defenders could only perform passive defenses and defenders could not interrupt or even destroy attackers. New models should be developed to include the defender's positive defense and the success rate of interdiction. Our model was defined on a network of nodes and arcs where each node was assumed to represent a local area of demand, as well as represent a potential position for a facility. In future work, we could choose the optimal position in a continuous network with no potential facilities. Regarding the partial optimization parameter, we could find a way to achieve self-adjustment rather than by manual adjustment.

**Author Contributions:** This manuscript was written by P.Y. under the supervision of Y.X. and Y.Z. The modeling, data analysis, and algorithm process were executed by S.Z. (Siyue Zhang) and P.Y.; S.Z. (Shenghan Zhou) and W.C. were responsible for the data acquisition and model design. All authors have read and agreed to the published version of the manuscript.

**Funding:** This work is supported by the National Natural Science Foundation of China (Grant Nos. 71871003, 71971013 and 71971009).

**Acknowledgments:** Thanks for the support of my friend: Meng You.

**Conflicts of Interest:** The authors declare no conflict of interest.

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
