# Peer review of "Dynamic Gaming Case of the R-Interdiction Median Problem with Fortification and an MILP-Based Solution Approach"

_sustainability, doi:10.3390/su12020581_

Round 1

Reviewer 1 Report

My main question is about the fit to the journal. What is the link with Sustainability? I think this question should be answered first.  I did not see the link reviewed the content supervisually.

More about the content:

it is unusual to use the past tense in a paper the problem has close relations to the Uncapicitated Facility Location Problem. Why is this not mentioned? Why is the extensive literature for those problems not used as a start for a solution method? line 141: should this be a small p? line 153: site line 161: should a be alfa? secton 3.2: the indices are not well formatted. In general: the formulas are not nicely formatted.

Reviewer 2 Report

Dear Authors 

The article can be accepted after a minor revision but it includes many factors. Please consider all the comments below: 

Keywords are not illustrating all the things you decide to do based on the title.  Conclusion and discussion should be expanded in detail.  You are not mentioning enough previous or related studies. Please add more with enough explanations.  The quality of the figures is not good enough. I feel you should mention more "the Sustainability"  Why your information is up to 2016?

Reviewer 3 Report

Presented methodology has a great potential in network security and linear programming field, so this topic in important for investigation. The strengths of this paper are: Relevant topic; Flow of the paper; Explanation of the methods and the results. However, the author(s) need to consider the following points as limitation or further scope for refining the paper:

- Introduction should be clearly stated targets. Then answer several questions: Why is the topic important (or why do you study on it)? What are your contributions? Why is to propose this particular model?

- Need to better highlight the novelty of study in the introduction. Page 1, line 104, the authors defined their contributions but this should be presented in more effective way. Add more contributions that are connected with managerial and social field.

- Figures 1, 3 and 4 are not cited in the text.

- I suggest authors to clearly summarize what specific advantages brings your approach.

- Literature review – The research gap and motivation should be clarified in this section. Based on literature review the authors should define the gap. Where is the gap? And you should clearly state why it is a gap? Once again, if you say that it is a gap, then try to build a case for the gap. As a part of this, you should add more recent papers with application of MILP model along with heuristic algorithms. Some relevant recent references are missing. For example, Biswas and Pal, 2019, presented interesting hybrid goal programing model and genetic algorithm in fuzzy environment (A fuzzy goal programming method to solve congestion management problem using genetic algorithm. Decision Making: Applications in Management and Engineering); Also, Barma et al, 2019, presented novel LP model with antlion optimization algorithm for multi-depot VRP (A 2-opt guided discrete antlion optimization algorithm for multi-depot vehicle routing problem. Decision Making: Applications in Management and Engineering).

- Discussion - The author(s) should present the advantage of the proposed method. The authors may present comparisons and results in section 4.2 in more effective way. Adding seven tables full of numbers and one figure is not useful. You should add deep discussion on the presented results. I must stress that results are good, but you should organize this section in more coherent way. This is essential section of your paper.

- Conclusion- Add future scope. Add limitations and advantages of proposed model.

Round 2

Reviewer 1 Report

See editor note.

Reviewer 3 Report

Most of the concerns have been properly addressed.